# Greensporone A, a Fungal Secondary Metabolite Suppressed Constitutively Activated AKT via ROS Generation and Induced Apoptosis in Leukemic Cell Lines

**DOI:** 10.3390/biom9040126

**Published:** 2019-03-29

**Authors:** Kirti S. Prabhu, Kodappully S. Siveen, Shilpa Kuttikrishnan, Anh Jochebeth, Tayyiba A. Ali, Noor R. Elareer, Ahmad Iskandarani, Abdul Quaiyoom Khan, Maysaloun Merhi, Said Dermime, Tamam El-Elimat, Nicholas H. Oberlies, Feras Q. Alali, Martin Steinhoff, Shahab Uddin

**Affiliations:** 1Translational Research Institute, Academic Health System, Hamad Medical Corporation, P.O. Box 3050, Doha, Qatar; KPrabhu@hamad.qa (K.S.P.); SSivaraman@hamad.qa (K.S.S.); SKuttikrishnan@hamad.qa (S.K.); AJochebeth@hamad.qa (A.J.); tayyiba1991@gmail.com (T.A.A.); NElareer@hamad.qa (N.R.E.); AIskandarani@hamad.qa (A.I.); AKhan42@hamad.qa (A.Q.K.); 2National Center for Cancer Care and Research, Hamad Medical Corporation, Doha 3050, Qatar; MMerhi@hamad.qa (M.M.); SDermime@hamad.qa (S.D.); 3Departent of Medicinal Chemistry and Pharmacognosy, Faculty of Pharmacy, Jordan University of Science and Technology, Irbid 22110, Jordan; tamamelimat@gmail.com; 4Department of Chemistry and Biochemistry, University of North Carolina at Greensboro, Greensboro, NC 27402, USA; Nicholas_Oberlies@uncg.edu; 5Qatar College of Pharmacy, Qatar University, Doha 3050, Qatar; feras.alali@qu.edu.qa; 6Department of Dermatology Venereology, Hamad Medical Corporation, Doha 3050, Qatar; MSteinhoff@hamad.qa; 7Weill Cornell-Medicine, Doha 3050, Qatar; 8Weill Cornell University, New York, NY 10065, United States

**Keywords:** AKT, apoptosis, cIAP, greensporone A, imatinib, reactive oxygen species

## Abstract

Greensporone A is a fungal secondary metabolite that has exhibited potential in vitro for anti-proliferative activity in vitro. We studied the anticancer activity of greensporone A in a panel of leukemic cell lines. Greensporone A-mediated inhibition of proliferation is found to be associated with the induction of apoptotic cell death. Greensporone A treatment of leukemic cells causes inactivation of constitutively activated AKT and its downstream targets, including members GSK3 and FOXO1, and causes downregulation of antiapoptotic genes such as Inhibitor of Apoptosis (IAPs) and Bcl-2. Furthermore, Bax, a proapoptotic member of the Bcl-2 family, was found to be upregulated in leukemic cell lines treated with greensporone A. Interestingly, gene silencing of AKT using AKT specific siRNA suppressed the expression of Bcl-2 with enhanced expression of Bax. Greensporone A-mediated increase in Bax/Bcl-2 ratio causes permeabilization of the mitochondrial membrane leading to the accumulation of cytochrome c in the cytoplasm. Greensporone A-induced cytochrome c accumulation causes the activation of caspase cascade and cleavage of its effector, poly(ADP-ribose) polymerase (PARP), leading to apoptosis. Greensporone A-mediated apoptosis in leukemic cells occurs through the generation of reactive oxygen species (ROS) due to depletion of glutathione (GSH) levels. Finally, greensporone A potentiated the anticancer activity of imatinib in leukemic cells. In summary, our study showed that greensporone A suppressed the growth of leukemic cells via induction of apoptotic cell death. The apoptotic cell death occurs by inhibition of AKT signaling and activation of the intrinsic apoptotic/caspase pathways. These results raise the possibility that greensporone A could be developed as a therapeutic agent for the treatment of leukemia and other hematological malignancies.

## 1. Introduction

Cancer is one of the prime causes of mortality and accounts for nearly 13% of deaths worldwide. The World Health Organization estimates that there might be 21.4 million cases of cancer and nearly 13.2 million deaths from cancer annually by 2030 [1]. Leukemia is a form of cancer in which white blood cells and their precursors dominate and differentiate abnormally, leading to suppression of production and functioning of healthy cells. Till date, the standard therapeutic strategies for cancer include chemotherapy, surgery, and radiation. However, the foremost problem in chemotherapy is the development of a cancer resistance mechanism, due to up-regulation of multi-drug resistance protein (MDR) and a decrease in the rate of apoptotic proteins [2]. Moreover, since the majority of chemotherapeutic treatments are associated with severe adverse effects, alternative forms of treatment using natural products are now considered as a promising area of research [3].

Compounds derived from natural resources have contributed significantly to the anticancer discovery process [4,5]. It has been estimated that out of the 246 small–molecules anticancer drugs that have been approved by the Food and Drug Administration (FDA) from the 1940s to 2014, 78% were those other than synthetic, with about 49% being either natural products or synthesized based on natural products [6]. Examples of anticancer drugs that are in use and derived from natural sources include actinomycin D, mitomycin C, bleomycin, etoposide, paclitaxel, docetaxel, and vincristine [5]. Fungal secondary metabolites have gained a lot attention as a source of novel cytotoxic scaffolds [4,7]. Current data estimates as many as 5 million species of fungi growing on earth, of which only a small percentage were studied for bioactive compounds, and an even smaller percentage has been explored for anticancer activity [8].

Structurally diverse classes of secondary metabolites such as alkaloids, diterpenes, triterpenes, and polyphenolic-type compounds have been reported to possess anticancer activity [9]. One such group of secondary metabolites is resorcyclic acid lactones (RAL) that belong to the family of benzannulated macrolides, produced by various fungal species, and are known to exhibit antibacterial, antifungal, antitumor, anti-inflammatory, antidiabetic, and antihypertensive activities [4,10,11,12,13]. As its name suggests, this group structurally consists of partially substituted β-resorcylic acid scaffold, linked to a 12- or 14-membered macro lactone moiety [14].

In our previously published study, 14 new RALs with promising cytotoxic activity against breast and colon cancer cell lines were isolated and characterized from a freshwater fungus that was collected from a stream in North Carolina [8]. Recently, our group published an article wherein we documented that greensporone C, one of the active RAL groups, induced apoptosis in leukemic cells through activation of the intrinsic/mitochondrial-mediated pathway of apoptosis [15]. As both secondary fungal metabolites were derived from that same fungus but possess different geometrical and substitution patterns on the 14-membered macrocyclic lactone ring, we would like to investigate whether greensporone A has any potential anti-leukemic effect against leukemic cell lines.

## 2. Materials and Methods

### 2.1. Isolation of Greensporone A from Aquatic Fungi

Greensporone A is a fungal secondary metabolite, which was isolated from an organic extract of an aquatic fungus *Halenospora* sp. (G87) collected from a stream running through the campus of the University of North Carolina at Greensboro, NC. After subjecting the organic extract and fractions to different purification procedures, greensporone A was isolated with >94% purity, as evidenced by UPLC. The compound was identified to have a molecular formula of C_19_H_21_ClO_6_ as determined by HRESIMS, while the structure of the compound was elucidated by extensive analysis of 1D and 2D NMR data [8].

### 2.2. Chemicals and Reagents

Caspase-9, caspase-8, caspase-3, cleaved-caspase-3, poly(ADP-ribose) polymerase (PARP), XIAP, cIAP-1, cIAP2, Bcl-2, Bcl-xL, and Bax were procured from Cell Signaling Technologies (Beverly, MA, USA) and the GAPDH antibody was procured from Santa Cruz Biotechnology, Inc. (Santa Cruz, CA, United States). Annexin V-FITC, propidium iodide staining solution, Hoechst33342Solution, BD Cytofix/Cytoperm Plus fixation and permeabilization solution kit(BD (Pharmingen San Jose, CA, USA). The Cell Counting Kit-8 (CCK-8) kit and N-acetyl cysteine (NAC) was obtained from Sigma-Aldrich (St. Louis, MO, United States). z-VAD-FMK was bought from Calbiochem (San Diego, CA, USA). CellROXGreen, MitoSOXRed, andThiolTracker Violet were purchased from Invitrogen (Waltham, MA, USA). Mitopotential kit was purchased from the EMD Millipore Corporation (Danvers, MA, USA).

### 2.3. Cell Culture

K562, U937, and AR230 leukemic cells were maintained in RPMI 1640 medium supplemented with fetal bovine serum (FBS, 10%), 100 U/ml penicillin, and 100 U/ml streptomycin at 37 °C in an atmosphere comprising of 5% CO_2_ [16].

### 2.4. Cell Proliferation Assay

Briefly, all leukemic cell lines were plated at a density of 1 × 10^4^ cells per well in 96-well microtiter plates and were treated with escalating concentrations of greensporone A for a period of 24 h. At the end of 24 h, CCK-8 solution was added to all the wells, plates were read at 450 nm, and percentage cell viability was calculated as described earlier [17].

### 2.5. Cell Cycle Analysis

Leukemic cells (K562 and U937) were treated with greensporone A as depicted in Figure 1C,D for a period of 24 h. At the end of treatment, cells were stained with Hoechst 33342 and cell cycle analysis was carried out using the flow cytometry BD LSRFortessa analyzer (BD Biosciences, NJ, United States) [18].

### 2.6. Annexin V/Propidium Iodide Dual Staining

Similar to cell cycle analysis, K562 and U937 cells were subjected to treatment with and without greensporone A for 24 h. Afterwards, cells were washed and then stained for 20 min with fluorescein-conjugated Annexin V and propidium iodide in 1X Annexin-binding buffer. Using flow cytometry technique the amount of cells that undergo changes after treatment with greensporone A were analyzed and expressed as a percentage, as mentioned earlier [19].

### 2.7. Cell Lysis and Immunoblotting

Proteins obtained from lysates of greensporone A-treated K562 and U937 cells were quantified with an ND-1000 spectrophotometer (NanoDrop Technologies, Thermo Scientific, United States). Protein lysates were resolved using SDS–PAGE, transferred to a polyvinylidenedifluoride (PVDF) membrane (Immobilon, Millipore, Billerica, MA, USA), then probed with various antibodies. Development and visualization of blots were done with ChemiDoc System (Amersham, Bio-Rad, Hercules, CA, USA) [20].

### 2.8. Measurement of Mitochondrial Membrane Potential

K562 and U937 leukemic cells were treated with varying doses of greensporone A for a time period of 24 h then subjected to analysis. As per manufacturer protocol, at the end of 24 h, cells were briefly centrifuged and suspended in 1X assay buffer. These cells were stained with 95 µL of mitochondrial potential dye and incubated for 20 min at 37 °C. At the end of 20 min, 5 µL of muse mitopotential 7-AAD reagent were added to all tubes and analyzed using a Muse cell analyzer [16].

### 2.9. Assay for Release of Cytochrome c

Greensporone A treated K562 and U937 leukemic cells were centrifuged at the end of 24 h treatment. Pellets obtained from centrifugation were resuspended in hypotonic buffer. As per the protocol mentioned earlier [21], cytosolic fractions were separated on 12% SDS–PAGE and immunoblotted with antibodies anti-cytochrome c and GAPDH [21].

### 2.10. Measurement of Mitochondrial Superoxide

Upon treatment with increasing doses of greensporone A for 24 h, leukemic cells K562 and U937 were washed with Hanks Balanced Salt Solution (HBSS) and then stained with MitoSOX Red Mitochondrial Superoxide Indicator, 5 µM (Invitrogen) in HBSS for a period of 20 min at 37 °C. Levels of mitochondrial superoxide were measured at Ex, 488 and Em, 575/526 by flow cytometry [22].

### 2.11. Measurement of Reactive Oxygen Species

K562 and U937 were treated with varying doses of greensporone A for 24 h. At the end of 24 h cells were washed, stained, and analyzed as described earlier by flow cytometry (Ex, 488; Em, 530/30) [23].

### 2.12. Measurement of Reduced Glutathione

At the end of 24 h treatment, leukemic cells with and without greensporone A cells were collected and washed using HBSS. HBSS with 10 µM ThiolTracker^TM^ Violet (Invitrogen, MA, United States) inserted was used as a staining solution to measure the amount of reduced glutathione using flow cytometry [24].

### 2.13. Gene Silencing of AKT Using Small Interference RNA

K562 cells (1 × 10^6^) were transfected using small interfering RNA (50 pmol and 100 pmol) to knock down the expression of AKT (S102758406, Qiagen, Germany). The cells were grown under standard conditions using an RPMI 1640 medium containing 10% fetal bovine serum and transfected using 4D-Nucleofactor system (Lonza), as per manufacturer protocol.

## 3. Statistical Analysis

Data obtained were expressed as mean ± standard deviation (SD). For comparison amongst experimental groups a paired student’s *t*-test was used. Statistical analysis and figure generation were done using GraphPad Prism (version 7.0 for Windows, GraphPad Software Inc., San Diego, CA, United States). * *P* ≤ 0.05, ** *P* ≤ 0.01, and *** *P* ≤ 0.001 were considered to be statistically significant.

## 4. Results 

### 4.1. Isolation and Characterization of Greensporone A from an Aquatic Fungus

Greensporone A was isolated as a colorless solid with a molecular formula of C_19_H_21_ClO_6_ as evidenced by HRESIMS. Using UPLC, the purity of the isolated compound was found to be >94%. The structure of the compound was elucidated using extensive analysis of 1D and 2D NMR data [8].

### 4.2. Effect of Greensporone A Treatment on Cell Proliferation and Apoptosis

To examine whether or not greensporone A had any effect on cell viability status of leukemic cell lines, we treated leukemic cell lines (K562, U937, and AR230) with and without greensporone A for 24 h then analyzed their cell proliferation capacity using CCK-8. As observed in Figure 1B, greensporone A significantly decreased cell viability in all three cell lines (IC_50_ ranging from 17 to 42 µM). A significant level of inhibition on cell viability was seen at 1 µM and above doses of greensporone A in all cell lines (Figure 1B).

In the next set of experiments, we sought to explore the effect of greensporone A on cell cycle and its potential to induce apoptosis in leukemic cell lines. Our data illustrate that in comparison to the control, there was a significant increase in SubG0/G1 phase in K562 and U937 cell lines treated with greensporone A for 24 h (Figure 1C–F). Annexin V/PI dual staining further verified the induction of apoptosis, as cells treated with 50 µM for 24 h showed an increased percentage of apoptosis in comparison to control cells (Figure 2A,B). To support these results, lysates obtained from treatment of leukemic cells with increasing doses of greensporone A for 24 h were probed with various antibodies for caspase-9, caspase-3, PARP, and GAPDH. Caspase-9 and caspase-3 are known to possess a critical role in apoptosis and, once activated, precursor forms are cleaved [25]. Western blot analysis revealed a decrease in expression of the precursor forms of caspase-9 and caspase-3, and an increase in their cleaved forms indicated the occurrence of apoptosis in leukemic cell lines following greensporone A treatment. In line with this, the intensity of PARP cleavage was found to increase in a dose-dependent manner (Figure 2C). Increased caspase-3 (Figure 2D) and PARP cleavage (Appendix A) activity in response to greensporone A treatment was further confirmed by flow cytometry. Phosphorylated H2AX (Figure 2E), a marker for double stranded breaks, was found to be significantly elevated in both K562 and U937 leukemic cell lines upon treatment with greensporone A (Figure 2F), implicating the existence of DNA double-stranded damage in greensporone A-induced apoptosis. To further confirm whether caspase activation is involved in greensporone A-mediated apoptosis, leukemic cells were pretreated with 20 μmol/l z-VAD-FMK, a pan-caspase inhibitor, for 1 h trailed by treatment with greensporone A (50 µM) for 24 h. After 24 h of treatment, cells were subjected to flow cytometry and western blot analysis. Greensporone A induced generation of apo fractions in cell cycle fraction analysis (Figure 3A). Annexin v/PI dual staining (Figure 3B) was abrogated by z-VAD-FMK pretreatment which was verified by western blot analysis, wherein the activity of caspases and PARPs were reversed on treatment with z-VAD-FMK (Figure 3C). In addition, greensporone A-induced caspase-3 activity (Figure 3D) as well as PARP cleavage activity (Figure 3E) were blocked by z-VAD-FMK pretreatment, confirming the involvement of caspases in greensporone A-induced apoptosis.

### 4.3. Treatment of Leukemic Cells with Greensporone A Suppressed the Activation of AKT and Its Associated Proteins

Previous findings by Hussain et al., 2017; Prabhu et al., 2018b; and Rivera et al., 2016 [15,26,27] have indicated the importance of AKT in regulating cell proliferation and its role in the PI3K-AKT pathway. Aberrant activation of such an important molecule is known to have oncogenic effects of AKT, as this is considered to be of prime AKT activation the levels of phosphorylated AKT were evaluated by western blot analysis. Our study found that greensporone A repressed the levels of phosphorylated AKT in a dose-dependent manner without altering levels of total AKT protein (Figure 4A). Forkhead Box-O (FOXO1) plays an important role in various cellular processes. Phosphorylation of FOXO1 by AKT leads to inhibition of transcriptional functions thereby contributing towards cell survival, growth, and proliferation [26,28,29]. The level of p-FOXO1 was found to be decreased in leukemic cell lines treated with greensporone A without affecting levels of FOXO1 (Figure 4A). Similarly, glycogen synthase kinase-3 (GSK3) regulates levels of glycogen synthase via phosphorylation and is known to be a downstream element of the PI3K-AKT pathway [29]. In normal cells, constitutively activated GSK3 is known to phosphorylate different molecular proteins such as cyclin D1 and c-myc, thereby promoting their inactivation or degradation. Upon phosphorylation activation of PI3K-AKT, GSK3 gets inactivated leading to increased expression of various tumorigenic proteins [30]. In our study, we observed that in treated cells the levels of p-GSK3 were suppressed in a concentration-dependent manner without any inhibition or decrease in the levels of total GSK3 (Figure 4A), thus suggesting that greensporone A targets AKT pathway in the generation of anti-proliferative effects in leukemic cells where AKT is found to be aberrantly activated.

Members of the IAP family are gaining recognition due to overexpression in many cancers and overpowering the apoptotic process [31,32]. Besides correlating with GSK3, several documented reports suggest a strong relationship between AKT and XIAP. Phosphorylation of XIAP halts the process of autoubiquitination and degradation of XIAP, and thereby causes an obstacle in caspase-induced apoptosis and supports resistance towards chemotherapeutic agents [33]. Various members of IAPs are also documented to be overexpressed upon transfection of constitutively activated AKT within them [34], signifying a strong relationship between AKT and IAP. Thus, in our study we investigated whether greensporone A-induced suppression of AKT has any link with IAP family members. Our data support the above facts as expression levels of XIAP and c-IAP1 were repressed in comparison to controls in both leukemic cell lines. Altogether, our data concludes the involvement of PI3K/AKT pathway and its downstream targets in inducing apoptosis following treatment with greensporone A (Figure 4B). This was further confirmed with experiments performed using a gene silencing approach with small interfering RNA (siRNA) of AKT. K562 cells were transfected with AKT specific siRNA using a nucleofactor system, as described in the methods. After transfection, levels of AKT, XIAP, caspase-3, Bax, and GAPDH were determined by immunoblotting against various antibodies. As depicted in Figure 4C, knocking out the AKT gene led to a reduction in expression levels of XIAP and procaspase-3 with an increase in levels of BAX.

### 4.4. Greensporone A Treatment Results in Activation of Caspase-8 and Dose-Dependent Decrease in Bcl-2 Protein Causing an Increase in Bax/Bcl-2 Ratio in Leukemic Cells

Treatment of leukemic cells K562 and U937 for 24 h with greensporone A decreased expression of precursor caspase-8 with a substantial increase in its cleaved form (Figure 5A). Caspase-8 is known to cleave Bid and, once cleaved, truncated BID gets translocated into mitochondria and activates Bax/Bak thereby causing damage to the mitochondrial membrane. Further, Raisova et al., 2001 [35] reported that lower levels of Bax/Bcl-2 ratio led to resistance, whereas a high ratio was linked to cells that are sensitive and undergo apoptosis in response to anticancer drugs. Increased expression of pro-survival protein, Bax, with the repressed activity of Bcl-2 was observed in K562 and U937 cells treated with greensporone A (Figure 5A). Densitometry analysis showed an association of high ratio of Bax/Bcl-2 with greensporone A induced apoptosis, and our findings are in substantiation with Raisova et al., 2001 and others (Figure 5B) [35,36,37].

### 4.5. Greensporone A-Mediated Activation of Mitochondrial Apoptotic Pathways in Leukemic Cells

Mitochondria is reported to play a vital role in the management of various cellular functions. Any form of intrinsic or extrinsic stress may cause mitochondrial membrane depolarization leading to the death of the cell through apoptosis or necrosis [38,39]. Thus, to investigate the involvement of mitochondria in greensporone A-induced cell death, leukemic cells (K562, U937) were treated with different doses of greensporone A for a period of 24 h and examined using a Muse analyzer. As observed in Figure 5C, treatment with greensporone A leads to loss of mitochondrial membrane potential. Depolarization in mitochondrial membrane potential is associated with cytochrome c release from mitochondria into cytosol during the process of apoptosis [40,41]. In concordance to these reports, in our study we observed that leukemic cells—upon treatment with greensporone A—lead to increased expression of cytosolic cytochrome c (Figure 5D). Released cytochrome c in the cytoplasm activates caspase cascades via forming an apoptosome complex in the presence of ATP and APAF-1 that finally causes apoptotic cell death [42,43].

### 4.6. Greensporone A-Mediated Generation of Reactive Oxygen Species (ROS) in Leukemic Cells

The contribution of free radicals is multifaceted in the progression of tumor cells and is an exceptional characteristic of cancer. Low levels of ROS act as mitogens thereby supporting cell proliferation and survival, whereas at intermediate concentrations, transient or permanent cell cycle arrest occurs. High concentration of ROS causes DNA damage thereby supporting cancer development [44]. Furthermore, ROS generation is considered a major factor in mitochondrial-dependent apoptosis. Therefore, in our study we explored the ROS inducing capability of greensporone A at the cellular and mitochondrial levels using flow cytometry. Leukemic cells K562 and U937 were exposed to treatment with escalating doses of greensporone A for 24 h. The significant upsurge in ROS cellular levels (Figure 6A), detected using cellROX; and mitochondrial levels (Figure 6B), detected by mitoSOXassay were observed in greensporone A treated leukemic cells [45]. Our findings suggest the involvement of greensporone A in the generation of ROS at both cellular and mitochondrial levels. This was further confirmed with pretreatment of leukemic cell lines with N-acetyl cysteine (NAC), an ROS scavenger, followed by greensporone A treatment for 24 h, wherein ROS generation induced by greensporone A was completely suppressed by NAC in K562 and U937 cells, at both cellular as well as mitochondrial levels (Figure 6C,D).

### 4.7. Effect of Greensporone A on Glutathione Leukemic Cell Lines

Till date association between glutathione (GSH) and apoptosis is not clearly understood, but GSH is still considered to have a critical role in cell survival. Low levels of GSH are known to support apoptosis whereas high levels have been conferred to oppose the process of apoptosis thereby providing therapeutic failure or resistance. Therefore, using flow cytometry techniques we assessed whether treatment of leukemic cells K562 and U937 with increasing doses of greensporone A for 24 h could reduce GSH level or not. As observed in Figure 6E, upon treatment with greensporone A, leukemic cells K562 and U937 caused significant depletion in GSH levels in a dose-dependent fashion. Furthermore, free radicals are scavenged by NAC thereby elevating levels of GSH [46]. We pretreated K562 and U937 leukemic cells with NAC followed by subsequent treatment with greensporone A and observed reversed activity induced by greensporone A in leukemic K562 and U937 cells (Figure 6F).

### 4.8. Greensporone A-Mediated ROS Generation Involved in Apoptotic Cell Death in Leukemic Cells

Our study provides evidence suggesting that apoptosis in leukemic cell lines occurs through ROS generation. Therefore, we wanted to investigate if ROS have any role in greensporone A-induced leukemic cell death. An increase in the SubG0/G1 fraction of cell cycle and rate of apoptosis was observed in greensporone A-treated leukemic cells. The activity of greensporone A was significantly reversed on pretreatment of leukemic cells with 10 mM NAC (Figure 7A–D, Appendix A). Western blot analysis also confirmed our findings wherein caspase activation and PARP cleavage induced by greensporone A was completely reversed (Figure 7E) suggesting that induction of apoptosis by greensporone A is governed by generation of ROS in leukemic cells.

### 4.9. Cytotoxic Activity of Imatinib and Greensporone A in Leukemic Cell Lines

To investigate the cytotoxic effects of imatinib and greensporone A as a monotherapy as well as in combination, K562 cells were exposed to treatment with imatinib (1 µM) and greensporone A (25 µM) alone and in combination for 48 h. Cell viability was assessed using the CCK-8 solution. Our data indicate a significant decrease in the rate of cell proliferation in the combination group compared to treatment alone (Figure 8A). Similar data were obtained wherein treatment with combination group resulted in enhanced positive Annexin V. A significant level of apoptosis was seen in the combination group compared to treatment alone (Figure 8B,C). Western blot data demonstrated a decrease in levels of XIAP and caspase-3, whereas an increase in levels of P-H2AX was observed in the combination group compared to the monotherapy group alone (Figure 8D).

Altogether, our data suggests that greensporone A induced apoptosis in leukemic cell lines via suppression of activated AKT and its downstream targets. In summation, greensporone A also mediated its apoptotic effect through induction of a mitochondrial-mediated pathway. A schematic diagram (Figure 9) illustrates the proposed apoptotic pathway induced by greensporone A in leukemic cell lines.

## 5. Discussion

Resorcyclic acid lactones belong to a group of fungal polyketide derivatives that are produced by a variety of fungal strains and are known to exhibit several pharmacological activities including anticancer properties [4]. We have investigated the anti-proliferative activity of two resorcyclic acid lactone greensporone A and greensporone C in leukemic cell lines. Greensporone A is a structurally related compound to greensporone C—both belonging to the resorcylic acid lactones class of natural compounds. However, greensporone C is a dechlorinated analogue of greensporone A. Another key difference was replacement of the C-5 ketone carbonyl in greensporone A by a methylene moiety in greensporone C. Both greensporone A and C showed inhibition of proliferation when tested against a panel of leukemic cell lines through induction of apoptotic cell death. However, greensporone C was found to be more potent in inducing cytotoxicity in comparison to greensporone A, this may be attributed to the presence of a chloro atom in greensporone A which decreases electron density due to the electron withdrawing effect and thus, less Van Der Waal interaction might be essential to binding. The chloro atom creates a steric hindrance, which might also negatively affect binding and will reduce lipophilicity, which affects solubility and partition coefficient. Till date, complete understanding of the molecular mechanism involved behind greensporone A-induced anti-proliferative activity has not been published.

In our present study, flow cytometric analysis of the DNA content in leukemic cells treated with greensporone A indicated an increase in the SubG0/G1 phase, suggesting that cells undergo apoptosis on treatment. This was further confirmed by Annexin V staining, wherein a dose-dependent increase in the percentage of apoptosis was observed. Caspase-3 activation followed by PARP cleavage results in apoptosis either through the intrinsic or extrinsic pathway. Leukemic cells treated with greensporone A exhibited decreased expression of procaspase-3 with subsequent increased expression of cleaved caspase-3 and PARP cleavage. As documented in various publications, caspase-3 activation is known to trigger PARP cleavage, which is considered a hallmark event in apoptosis [47]. We also used broad-range pan-caspase inhibitor z-VAD-FMK, which inhibited PARP cleavage induced by greensporone A demonstrating the existence of activation of caspase cascade resulting in apoptosis.

PI3K/AKT is known to be involved in various biological functions and is also documented to be the most frequently deregulated pathway in cancer [48]. Deregulation of this pathway causes amplification and mutation of genes encoding PI3K and AKT. Targeting the PI3K/AKT pathway and its downstream targets like FOXO1 and GSK3 appears to be an attractive approach to treat cancers that are known to have aberrant AKT activation [49]. High levels of constitutive AKT has been associated with a short survival rate in cancer patients [50,51]. Our study revealed that the levels of phosphorylated AKT were downregulated in leukemic cell lines on treatment with greensporone A for 24 h without affecting the levels of AKT proteins. We also showed that inactivation of AKT was accompanied by inactivation of its downstream targets like FOXO1 and GSK3 in leukemic cells treated with greensporone A. The role of AKT is very diverse and it has found to be connected with regulations of various members of IAP family [52,53,54]. In this study, we also observed that various members of IAP families were downregulated in a dose-dependent manner upon treatment of leukemic cell lines with greensporone A. These findings indicate that greensporone A induces its pro-apoptotic effect through downregulation of activated AKT and its downstream targets.

Apoptosis occurs in various organisms via an extrinsic or intrinsic pathway through mitochondria. Bax, a pro-apoptotic molecule; and Bcl-2, an anti-apoptotic molecule, act opposingly in mitochondria. Most anticancer agents induce their activity by upregulating Bax and downregulating Bcl-2 thereby supporting apoptosis. Similar to our previous findings, the present study also demonstrated upregulation in levels of Bax with a concomitant decrease in levels of Bcl-2 in leukemic cell lines treated with greensporone A [15]. Altered Bax/Bcl-2 ratio caused a disturbance in mitochondrial potential resulting in the release of cytochrome c into the cytosol. Cytochrome c further activates procaspase-3 via Apaf-1/caspase-9. Activated procaspase further activates caspase cascade and PARP cleavage resulting in apoptosis [55]. Our findings also show that greensporone A-induced caspase cascade activation together with subsequent PARP cleavage signifying greensporone A-mediated apoptosis occurs via the intrinsic pathway.

Either through the process of intracellular oxygen metabolism or via extracellular environment, ROS are known to be generated. Elevated levels of ROS generate oxidative stress, whereas upon oxidation, macromolecules cause cellular damage to organelles and genes resulting in cell death by apoptosis. Several disorders are known to arise due to oxidative stress, and this oxidative stress plays a dual role in the progression as well as regression of cancer [56]. Several agents like α-tocopherol and ascorbic acid are known to induce apoptosis through ROS generation. Our data depicted a dose-dependent elevation in ROS with the subsequent decrease in levels of GSH. Furthermore, our data also supported findings that greensporone A-induced proapoptotic effects are ROS dependent, as pre-exposure of leukemic cells with ROS scavenger abolished caspase activation and PARP activation induced by greensporone A in leukemic cell lines.

## 6. Conclusions

Our study validates that greensporone A-induced apoptosis acts through inactivation of AKT and its downstream targets. Downregulation of Bcl-2 with an increased level of Bax resulted in the collapse of mitochondrial membrane potential leading to the release of cytochrome c, which further supported caspase cascade activation and PARP cleavage leading to activation of the mitochondrial-mediated pathway in leukemic cells. The anti-proliferative potential of greensporone A was mediated via ROS generation. Pretreatment of leukemic cells with NAC abolished both caspase cascade activation and GSH depletion induced by greensporone A. Altogether, our study shows that greensporone A modulates apoptotic response in human leukemic cell lines, thereby raising its possibility to be used alone, or in combination with conventional chemotherapeutic agents like imatinib, for treatment of hematological malignancies.

## Figures and Tables

**Figure 1 biomolecules-09-00126-f001:**
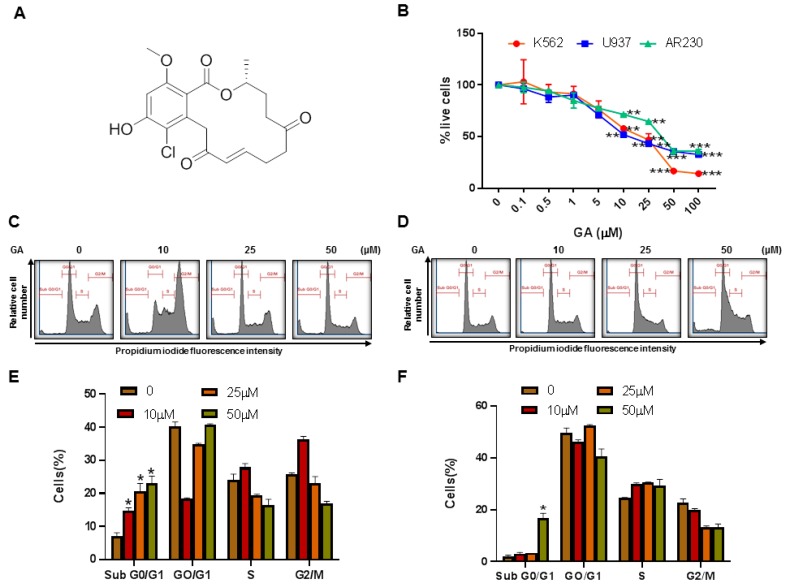
Effects of greensporone A (GA)–on cell proliferation and cell cycle. (**A**) Molecular structure of Greensporone A. (**B**) MTT assay was used to measure cell viability as mentioned in Section 2. Cell cycle fraction analysis of cells in response to GA. (**C**) K562 and (**D**) U937 cells were treated with GA, as indicated, and analyzed by flow cytometry. GA significantly enhanced SubG0 fraction in (**E**) K562 and (**F**) U937. The graph displays the mean ± SD of three independent experiments. * *P* < 0.05, ** *P* < 0.01, *** *P* < 0.001.

**Figure 2 biomolecules-09-00126-f002:**
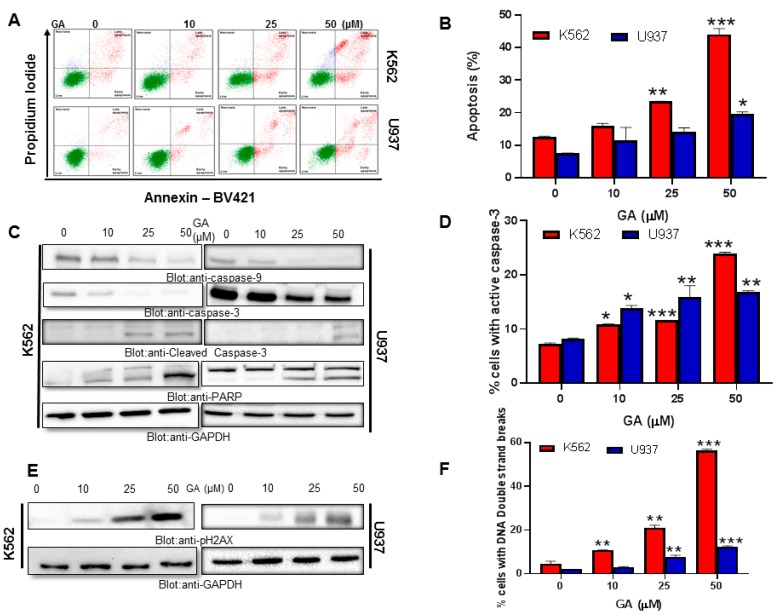
Greensporone A induces apoptosis in leukemic cells. K562 and U937 cells were (**A**) treated with GA in a dose-dependent manner and (**B**) analyzed by flow cytometry. Greensporone A-mediated caspase cascade activation in K562 and U937 cells as analyzed by (**C**) western blot and (**D**) flow cytometry, followed by DNA double strand breakage analyzed by the same methods, (**E**) and (**F**), respectively, in K562 and U937 cells. The graph displays the mean ± SD of three independent experiments. * *P* < 0.05, ** *P* < 0.01, *** *P* < 0.001.

**Figure 3 biomolecules-09-00126-f003:**
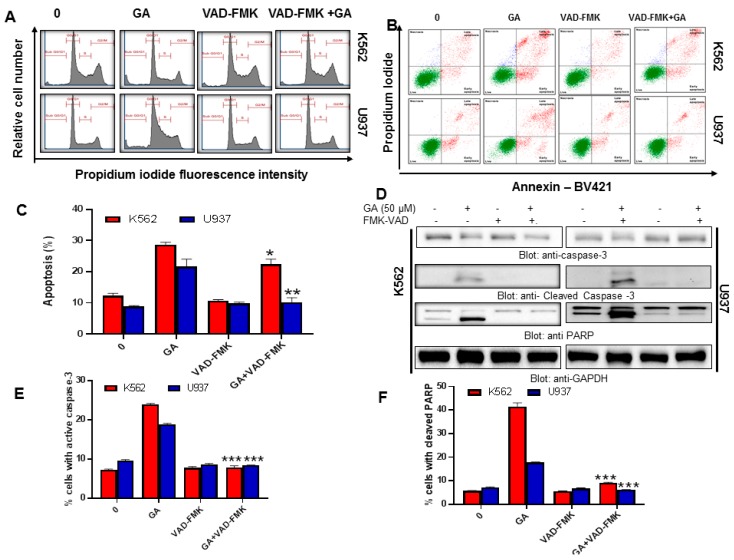
z-VAD-FMK reversed greensporone A-induced cell cycle and caspase activation in leukemic cells. Leukemic cells K562 and U937 cells were treated with greensporone A and z-VAD-FMK alone and in combination for 24 h. (**A**,**B**) z-VAD-FMK reversed greensporone A-induced increased SubG0 phase and apoptosis, respectively. (**C**–**E**) Caspase activation induced by greensporone A in K562 and U937 leukemic cells were reversed by z-VAD-FMK as analyzed by western blot and flow cytometry. The graph displays the mean ± SD of three independent experiments. * *P* < 0.05, ** *P* < 0.01, *** *P* < 0.001.

**Figure 4 biomolecules-09-00126-f004:**
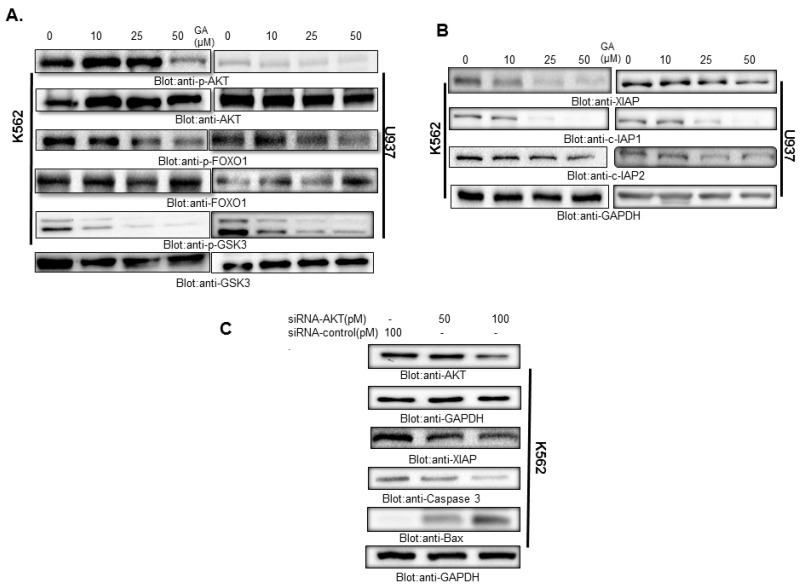
Leukemic cells treated with greensporone A suppressed the activation of AKT and its associated proteins. (**A**,**B**) K562 and U937 cells were treated with increasing doses of greensporone A for 24 h, as indicated. After cell lysis, equal amounts of proteins were separated by SDS–PAGE, transferred to polyvinylidenedifluoride (PVDF) membrane, and immunoblotted with antibodies of p-AKT, AKT, p-FOXO1, FOXO1, p-GSK3, GSK3, XIAP, c-IAP1, c-IAP2, and GAPDH as indicated. Gene silencing using siRNA of AKT in K562 cell line. (**C**) K562 cells were transfected with scrambled siRNA and AKT siRNA (50 and 100 picomolar((pM)). After 48 h, cells were lysed, and proteins were immunoblotted with antibodies against AKT, XIAP, Caspase 3, Bax, and GAPDH.

**Figure 5 biomolecules-09-00126-f005:**
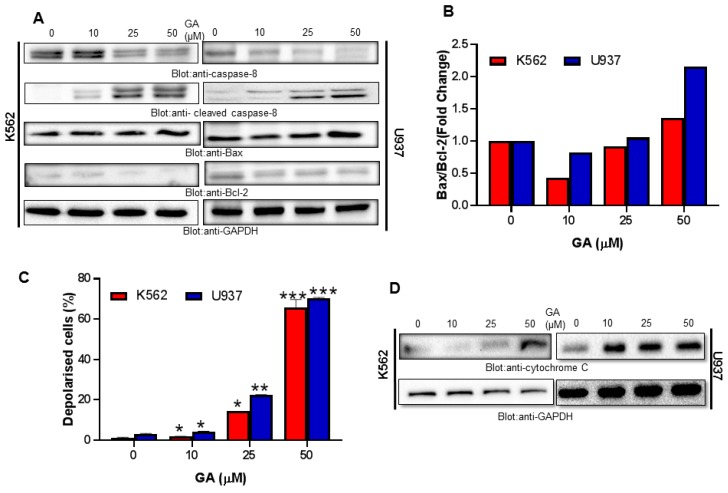
Greensporone A-induced mitochondrial signaling pathways in leukemic cells. Greensporone A treatment causes alteration in Bcl-2 expression. (**A**) K562 and U937 cells were treated with increasing doses of greensporone A for 24 h, as indicated. After cell lysis, equal amounts of proteins were separated by SDS–PAGE, transferred to PVDF membrane, and immunoblotted with antibodies against caspase-8, cleaved caspase-8, Bax, Bcl-2, and GAPDH. (**B**) Data obtained from immunoblot analysis of Bax and Bcl-2 in K562 and U937 were used to evaluate effects of GA on Bax/Bcl-2 ratio. Densitometric analysis of Bax and Bcl-2 bands was performed using AlphaImager Software (San Leandro, CA, USA), and data (relative density normalized to b-actin) were plotted as Bax / Bcl-2 ratio. Treatment with greensporone A caused loss of mitochondrial membrane potential in leukemic cells. (**C**) K562 and U937 cells were treated with increasing concentrations of greensporone A for 24 h and analyzed by Muse analyzer described in Section 2. The graph displays the mean ± SD of three independent experiments (* *P* < 0.05, ** *P* < 0.01, and *** *P* < 0.001). Greensporone A-induced the release of cytochrome c. (**D**) K562 and U937 cells were treated in the presence and absence of greensporone A for 24 h. Cytoplasmic fraction was isolated as described in Section 2. Cell extracts were separated on SDS-PAGE, transferred to PVDF membrane, and immunoblotted with an antibody against cytochrome c and GAPDH.

**Figure 6 biomolecules-09-00126-f006:**
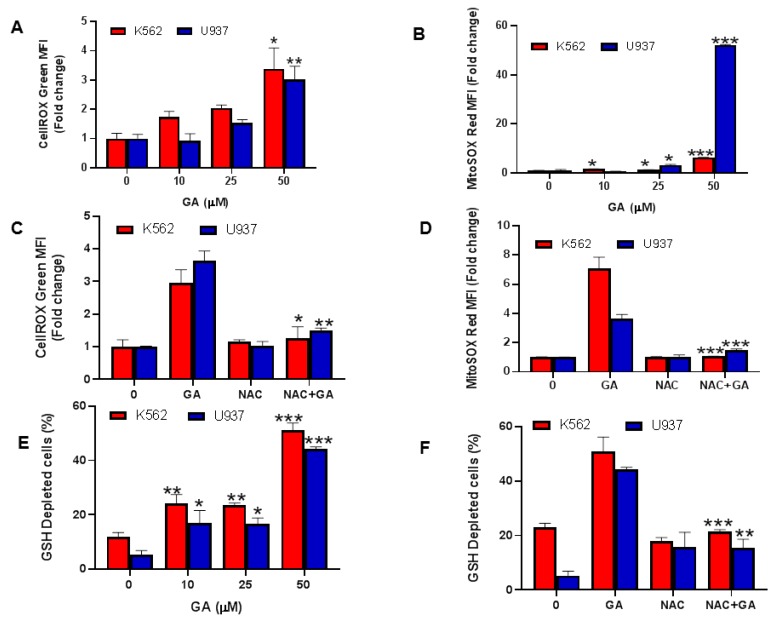
Greensporone A-mediated generation of reactive oxygen species (ROS) in leukemic cells. K562 and U937 were treated with greensporone A for 24 h. (**A**–**D**) Cellrox and mitoSOX assays were performed to evaluate the level of ROS by flow cytometry as described in Section 2. (**E**,**F**) Glutathione level was determined by flow cytometry. K562 and U937 cells were treated with greensporone A for 24 h and glutathione level was determined using a ThioTracker assay kit. The graph displays the mean ± SD (standard deviation) fold change release of ROS of three experiments (* *P* < 0.05, ** *P* < 0.01, *** *P* < 0.001).

**Figure 7 biomolecules-09-00126-f007:**
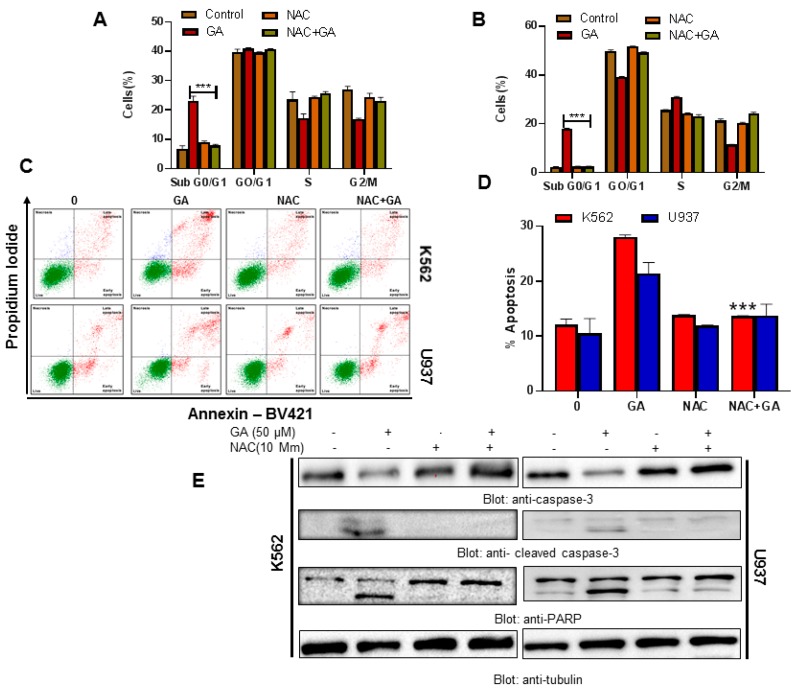
Greensporone A-mediated ROS generation involved in apoptotic cell death in leukemic cells. (**A**,**B**) N-acetyl cysteine (NAC) pre-treated leukemic cells abrogated the greensporone A-induced increase in SubG0 fraction in K562 and U937 cells. K562 and U937 cells were pretreated with 10 mM NAC followed by 50 µM GA for 24 h, and cell cycle fraction was measured by flow cytometry. (**C**,**D**) K562 and U937 cells were pretreated with 10 mM NAC followed by 50 µM GA for 24 h, and apoptosis was measured by staining with fluorescein-conjugated Annexin V and propidium iodide (PI) and analyzed by flow cytometry. The graph displays the mean ± SD of three independent experiments (* *P* < 0.05, ** *P* < 0.01 and *** *P* < 0.001). NAC pre-treated leukemic cells prevented greensporone A-mediated activation of caspases. (**E**) K562 and U937 cells were pretreated with 10 mM NAC then subsequently treated with 50 µM GA as indicated for 24 h. Lysed cell extracts were separated on SDS-PAGE, transferred to PVDF membrane, and immunoblotted with an antibody against procaspase-3, cleaved caspase-3, PARP, and tubulin.

**Figure 8 biomolecules-09-00126-f008:**
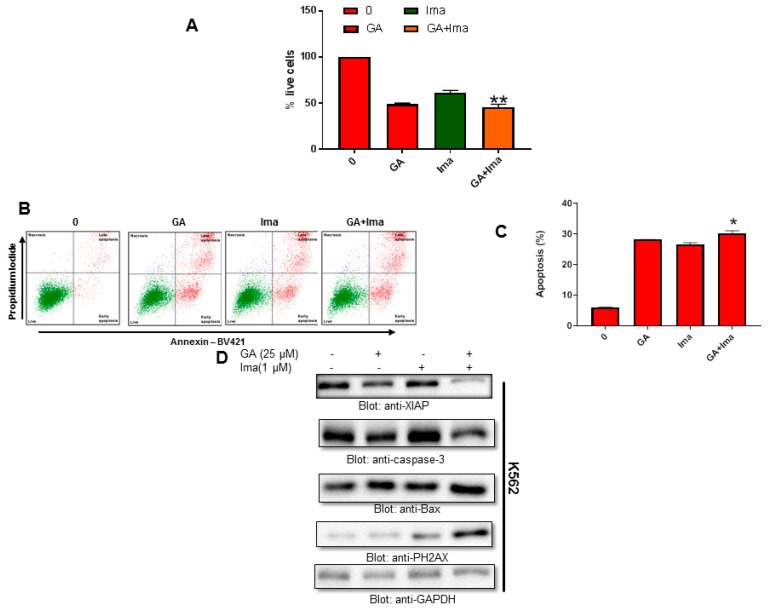
Cytotoxic activity of imatinib and greensporone A in leukemic cell lines. (**A**–**C**) K562 cells were treated with greensporone A (25 μM) and imatinib (1 μM) alone and in combination, and analyzed for cell proliferation and apoptosis by flow cytometry. The graph displays the mean ± SD of three independent experiments (* *P* < 0.05, ** *P* < 0.01 and *** *P* < 0.001). (**D**) K562 leukemic cells were treated with greensporone A (25 μM) and imatinib (1 μM) for 48 h and lysed cell extracts were separated on SDS-PAGE, transferred to PVDF membrane, and immunoblotted with an antibody against XIAP, procaspase-3, BAX, PH2AX, and GAPDH.

**Figure 9 biomolecules-09-00126-f009:**
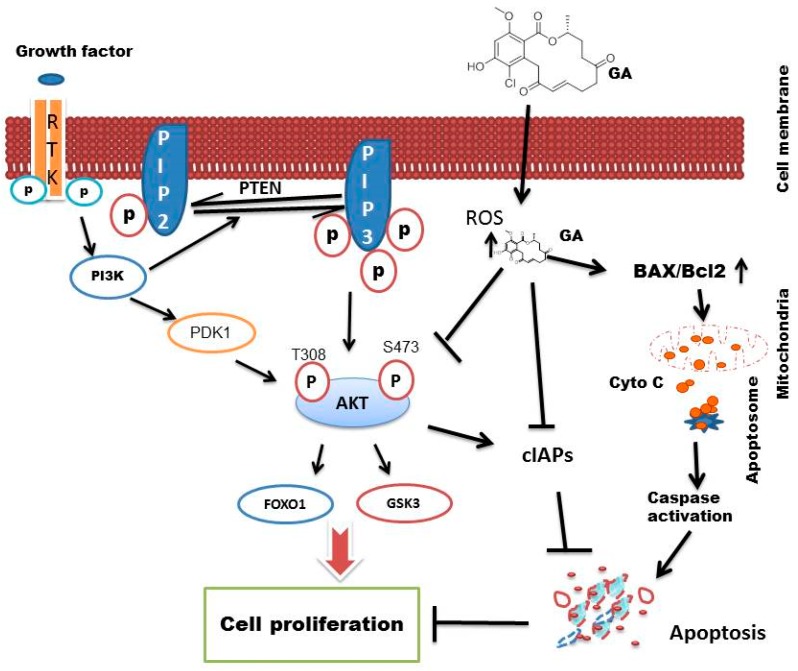
Schematic diagram depicting greensporone A induced apoptosis in leukemic cell lines.

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
