# Peer review of "Greensporone A, a Fungal Secondary Metabolite Suppressed Constitutively Activated AKT via ROS Generation and Induced Apoptosis in Leukemic Cell Lines"

_biomolecules, 2019, doi:10.3390/biom9040126_

Round 1

Reviewer 1 Report

The paper by Prabhu et al. reports the assessment of the antiproliferative activity of a natural compound and a study of its mechanism of  action. The experiments were conducted  following standard procedures and the results highlight the pathways that are affected by the compound.

Therefore the paper is of interest and is worth of publication.

Comments:

The IC50s of the antiproliferative activity should be measured and reported. This will allow a comparison with standard compounds and thus furnish information on the actual potency of the compound.

The Authors have published a similar study (ref. 15) on a strictly related compound, greensporone C, that belongs to the same class of natural compounds. Readers would expect a discussion on the similarities and differences of the behaviour of the two compounds, and, if possible, a discussion of structure-activity relationship 

Author Response

The paper by Prabhu et al. reports the assessment of the antiproliferative activity of a natural compound and a study of its mechanism of action. The experiments were conducted following standard procedures and the results highlight the pathways that are affected by the compound.

Therefore the paper is of interest and is worth of publication

Authors Response (AS):  We appreciated the reviewer for positive comments to our study and for mentioning that “paper is of interest and is worth of publication”.

Comments:

The IC50s of the antiproliferative activity should be measured and reported. This will allow a comparison with standard compounds and thus furnish information on the actual potency of the compound

AS:  The IC 50 of greensporon A in leukemic cell lines ranges from 17-42µM. We have incorporated this information in the revised manuscript line 177, Page 5

The Authors have published a similar study (ref. 15) on a strictly related compound, greensporone C, that belongs to the same class of natural compounds. Readers would expect a discussion on the similarities and differences of the behavior of the two compounds, and, if possible, a discussion of the structure-activity relationship.

AS: Greensporone A is structurally related compound to greensporone C; both belonging to resorcylic acid lactones class of natural compounds. However, greensporone C is a dechlorinated analog of greensporone A. Another key difference was replacement of the C-5 ketone carbonyl in greensporone A by a methylene moiety in greensporone C.  Although both compounds showed cytotoxic response towards leukemic cells however the greensporone c is more potent than greensporon A. Both greensporone A and C showed inhibition of proliferation when tested against a panel of leukemic cell lines through induction of apoptotic cell death. However, greensporone C was found to be more potent in inducing cytotoxicity in comparison to greensporone A, this may be attributed to the presence of chloro atom in greensporone A which decreases electron density due to electron withdrawing effect and thus, less Van Der Waal interaction which might be essential to binding. Chloro atom creates a steric hindrance which also might negatively affect binding and will reduce lipophilicity which affects solubility and partition coefficient. This paragraph has been introduced under discussion (line 407, Pg15).

Reviewer 2 Report

Prabhu et al have investigated anticancer activity of a secondary metabolite of fungus, Greensporone A, on a panel of leukemic cell lines. The authors used numerous biochemical and molecular techniques to measure proliferation, apoptosis, oxidative stress in response to  Greensporone A. The studies showed that Greensporone A causes dose-dependent inhibition of cell viability via induction of apoptosis. The Greensporone A induces cytotoxic effects by inactivation of AKT and its associated signaling pathways. The apoptotic cell death is  triggered by involving mitochondrial and caspase-mediated pathways.  Interestingly, Greensporone A potentiated the anticancer activity of imatinib in leukemic cells suggesting that Greensporone A could be developed as a therapeutic agent for the treatment of leukemia and other hematological. This is a novel and comprehensive study with lots of experimental data. The hypothesis of this study is well supported by data and over it is a well-presented study.  
Comments/suggestions.
1.    Imatinib and Greensporone A  combination of data information is not mentioned  in the abstract
2.    A schematic diagram of findings can be summarized in a form of figure.

Author Response

Prabhu et al have investigated anticancer activity of a secondary metabolite of fungus, Greensporone A, on a panel of leukemic cell lines. The authors used numerous biochemical and molecular techniques to measure proliferation, apoptosis, oxidative stress in response to  Greensporone A. The studies showed that Greensporone A causes dose-dependent inhibition of cell viability via induction of apoptosis. The Greensporone A induces cytotoxic effects by inactivation of AKT and its associated signaling pathways. The apoptotic cell death is triggered by involving mitochondrial and caspase-mediated pathways.  Interestingly, Greensporone A potentiated the anticancer activity of imatinib in leukemic cells suggesting that Greensporone A could be developed as a therapeutic agent for the treatment of leukemia and other hematological. This is a novel and comprehensive study with lots of experimental data. The hypothesis of this study is well supported by data and over it is a well-presented study.   

We are very thankful to the reviewer for mention that “This is a novel and comprehensive study with lots of experimental data. The hypothesis of this study is well supported by data and over it is a well-presented study”. 

Comments/suggestions.

1.     Imatinib and Greensporone A  combination of data information is not mentioned  in the abstract

AS: The imatinib and Greensporone A  combination of data information is mentioned in the abstract (line 40, Page 1).

2.     A schematic diagram of findings can be summarized in a form of the figure.

AS: We have added a new schematic diagram depicting the action of greensporone A in the revised manuscript (line 396, Page 14)